# The Protection, Challenge, and Prospect of Anti-Oxidation Coating on the Surface of Niobium Alloy

**Xu Zhang** [1], **Tao Fu** [1], **Kunkun Cui** [1], **Yingyi Zhang** [1,*], **Fuqiang Shen** [1], **Jie Wang** [1], **Laihao Yu** [1] and **Haobo Mao** [2]

1    School of Metallurgical Engineering, Anhui University of Technology, Maanshan 243002, China;
zx13013111171@163.com (X.Z.); ahgydxtaofu@163.com (T.F.); 15613581810@163.com (K.C.);
sfq19556630201@126.com (F.S.); wangjiemaster0101@outlook.com (J.W.); aa1120407@126.com (L.Y.)
2    School of Civil Engineering and Architecture, Anhui University of Technology, Maanshan 243002, China;
L1499923420@163.com
*    Correspondence: zhangyingyi@cqu.edu.cn; Tel.: +86-173-7507-6451

**Abstract:** Niobium (Nb)-based alloys have been extensively used in the aerospace field owing to their excellent high-temperature mechanical properties. However, the inferior oxidation resistance severely limits the application of Nb-based alloys in a high-temperature, oxygen-enriched environment. Related scholars have extensively studied the oxidation protection of niobium alloy and pointed out that surface coating technology is ideal for solving this problem. Based on the different preparation methods of Nb-based alloys' surface coatings, this article summarizes the relevant research of domestic and foreign scholars in the past 30 years, including the slurry sintering method (SS), suspension plasma spraying method (SPS), and halide activated pack cementation method (HAPC), etc. The growth mechanism and micromorphology of the coatings access by different preparation methods are evaluated. In addition, the advantages and disadvantages of various coating oxidation characteristics and coating preparation approaches are summarized. Finally, the coating's oxidation behavior and failure mechanism are summarized and analyzed, aiming to provide valuable research references in related fields.

**Keywords:** niobium alloy; oxidation resistance; surface coating; growth mechanism; oxidation behavior

## 1. Introduction

With the human need for space exploration, the development of hypersonic vehicles has attracted wide attention worldwide [1,2]. Due to the nature of long-term hypersonic cruises and the flight of hypersonic aircraft back and forth between the atmosphere and atmospheric reentry [3], the aircraft must face extremely harsh environments, producing high dynamic pressure and aerodynamic heating effects [4,5]. The high-temperature structural materials need to withstand extreme thermal and mechanical loads [6], resulting in large temperature gradients and thermal stresses inside the material, thereby significantly reducing the cycle life of the components. Especially critical parts or components include aircraft nose cones, sharp leading edges, nozzle openings, hot ends of engines [7], etc. Accordingly, the thermal development and high-temperature oxidation resistance of high-temperature structural materials are increasingly required. Traditional steel materials, aluminum alloys, and titanium alloys can no longer meet the extreme environmental requirements of hypersonic aircraft [8]. Niobium and its alloys have become a critical applicant material for high-temperature structural parts in the aerospace and nuclear industries due to their high melting point [9], moderate density, excellent high-temperature strength, and good processability. Niobium-based alloys are expected to replace nickel-based materials and become critical structural materials in the aerospace field by the end of the 21st century [10,11]. Nonetheless, the oxidation resistance of niobium-based alloys is lacking [12], and severe pulverization will occur when exposed to air above

500 °C for a short time, severely restricting its application in high-temperature, oxygen-rich environments. At present, the commonly used methods to inhibit the occurrence of this kind of oxidation include alloying and surface coating technology [13]. Although alloying can improve the corrosion resistance of niobium-based alloys in high temperature and oxygen-enriched environments to a certain extent, this measure often seriously affects the physical properties of the base alloy itself. Surface coating technology is the most effective method for enhancing the oxidation resistance of niobium-based alloys while ensuring the substrate's physical properties [13]. The arrangement of high-temperature, oxidation-resistant defensive coatings on the surface of niobium alloys has developed into a current research hotspot.

At present, there are many reports on the surface coating and oxidation protection of niobium and its alloys, but there are few studies on the growth mechanism and oxidation behavior of the surface coating. This article reviews the main preparation methods of niobium and its alloy surface coatings in recent years (such as the slurry sintering method (SS), suspension plasma spraying method (SPS), halide activated filling cementing method (HAPC), etc.). The latest research status of high-temperature, oxidation-resistant coatings on niobium-based alloys is discussed, and the advantages and disadvantages of various preparation methods are analyzed and summarized. The microscopic morphology, phase composition, and oxidation resistance of coatings prepared by different methods are compared and analyzed. The growth mechanism and oxidation behavior of various coatings are analyzed and summarized. At the same time, the future development direction of niobium and its alloy surface coatings is put forward with the purpose of adding valuable summaries for researchers on this ground.

## 2. Anti-Oxidation Coating on Niobium Alloys

The comprehensive properties of niobium alloy surface coatings are different due to the different preparation methods. In the following summary, different methods for preparing niobium alloy surface coatings will be described in detail, and the oxidation resistance performance of coatings prepared by different methods will be comprehensively compared.

### 2.1. Slurry Sintering Method

The slurry sintering method is the most commonly used coating preparation method on the surface of niobium alloys, and the process flow is shown in Figure 1. Firstly, the coating slurry is uniformly mixed with the components of the coating and the binder in proportion, to prepare the coating slurry [14,15], and the slurry is applied to the surface of the substrate by brushing, dipping, or spraying. Then it is solidified by pressurization and heating, and sintered in a vacuum or atmosphere furnace. Finally, a coating is formed on the face of the substrate. The process conditions, composition thickness [16], and the oxidation characteristics of the SS coatings on Nb-based alloys are summarized in Table 1. It can be perceived that the thickness of SS coatings was between 150 and 300 μm under the sintering temperature of 1200 to 1500 °C for 1–4 h in a vacuum or Ar atmosphere.

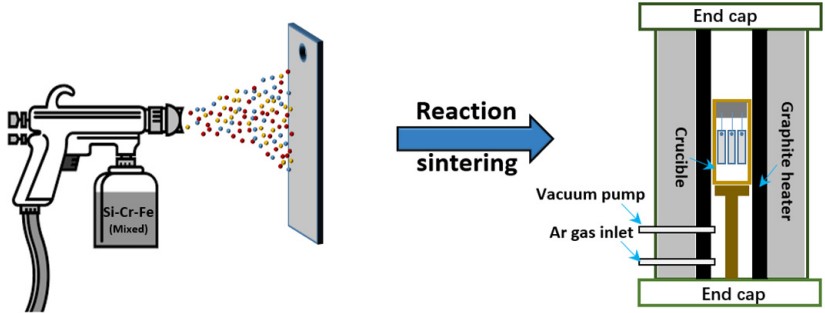

**Figure 1.** Flow chart of coating preparation by the slurry method.

**Table 1.** Summary for preparation and oxidation resistance of SS coatings on Nb-based alloys.

| Substrate | Slurry Composition (wt.%) | Process Conditions | Coating Composition and Thickness (μm) | | Oxidation System | | Oxidation Products | Quality Change (mg/cm$^2$) | References |
|---|---|---|---|---|---|---|---|---|---|
| | | | Outer Layer | Interface Layer | Oxidation Temperature (°C) | Oxidation Time (h) | | | |
| C-103 | 60Si-15Fe-20Cr-5NaF | 1400 °C/2 h vacuum | NbSi$_2$ Cr$_3$Si Fe$_3$Si$_2$ (200) | Nb$_5$Si$_3$ (30) | 1300 | 2.5 | Nb$_2$O$_5$ SiO$_2$ Fe$_2$O$_3$ | 1.5 | [17] |
| R512E | 20Si-35Fe-35Cr-10NaF | 1250 °C/4 h vacuum | MoSi$_2$ M$_5$Si$_3$ (135) | Nb$_5$Si$_3$ (15) | 1200 | 2 | Nb$_2$O$_5$ Cr$_2$O$_3$ SiO$_2$ CrNbO$_4$ | 1.8 | [18] |
| Nb521 | 45Mo-45Si-10NaF | 1500 °C/1 h Ar | NbSi$_2$ MoSi$_2$ (110) | Nb$_5$Si$_3$ (25) | 1700 | 25 | SiO$_2$ MoSi$_2$ | −9.5 | [19] |
| Nb-Si | 60Si-15Fe-20Cr-5NaF | 1400 °C/2 h vacuum | NbSi$_2$ Nb$_4$Si$_5$CrFe$_3$ Fe$_4$Nb$_4$Si$_7$ (250) | Nb$_5$Si$_3$ (50) | 1400 | 6.5 | Nb$_2$O$_5$ SiO$_2$ | 3.5 | [20] |
| Nb-Si-Ti | 20Fe-20Cr-50Si-10NaF | 1400 °C/2 h vacuum | NbSi$_2$ (Fe, Cr)$_3$Si$_2$ (170) | Nb$_5$Si$_3$ (20) | 1400 | 7 | Nb$_2$O$_5$ SiO$_2$ | 1.9 | [21] |

The typical surface and corresponding cross-sectional morphologies of SS coatings are shown in Figure 2. It can be noticed that the coating surface was relatively rough with dozens of holes and cracks. This was caused by the considerable particle size of the mixture, uneven mixing, and volatilization of the solvent and binder during the sintering process, as shown in Figure 2a,b. It is worth noting that Xiao et al. used smaller material particles for sintering and obtained a lower surface roughness, and the coating surface was relatively uniform and dense without apparent defects, as shown in Figure 2c. It can be observed from the cross-sectional images that the coating consisted of the outermost layer (NbSi$_2$), the intermediate layer, and the internal interdiffusion layer (IZD). In addition, as there was a mismatch of thermal expansion coefficient between the substrate and the coating during the sintering process, a small number of longitudinal cracks were observed inside the coating [22]. However, the IZD area, where the coating and the substrate were connected, was heavy and uniform, revealing that an excellent metallurgical bond was achieved between the coating and the substrate [23], as shown in Figure 2d–f. Han et al. [24] systematically explained the delamination phenomenon of such coatings. They believed that the growth process of the coating was firstly combined by chemical adsorption and physical adsorption, and then polar groups and substrates diffused between them at high temperatures. Finally, a firm and dense interwoven network coating layer was formed at the interface.

Relevant scholars have researched the oxidation resistance of the coating. After oxidation in the series of 1200–1700 °C for 2–25 h, the mass transition per unit area of the coating was within 10 mg·cm$^{-2}$, as shown in Table 1. The typical surface and cross-section microstructures of oxidized coatings are shown in Figure 3. It can be perceived that an oxide layer was formed on the oxidized coating surface, and the oxide layer consisted of SiO$_2$ and Nb$_2$O$_5$. Compared with before oxidation, the coating surface was relatively smooth at the initial oxidation stage [25]. This was owing to the formation of SiO$_2$ with a certain fluidity during the oxidation process, which filled up weaknesses such as cracks and gaps on the coating surface, to a certain extent. The inner coating was quite dense without apparent defects [26], as shown in Figure 3b,e. As the oxidation reaction progressed, the thickness of the oxide layer grew gradually. Several longitudinal cracks throughout the coating to the substrate were observed at the coating cross-section. This was caused by the large amount of volatile NbO$_2$ produced during the oxidation process, as shown in Figure 3c. At

the same time, a large number of longitudinal cracks throughout the entire cross-section were observed in the coating, and the same composition as the oxide layer was detected in this area, which shows that the coating could no longer provide adequate protection for the substrate, as shown in Figure 3f. Thermal diffusion referred to the spontaneous transition of matter to an equilibrium state in a high temperature and closed environment, significantly affected by temperature. Due to the principle of thermal diffusion [27], the thickness of the interface layer gradually increased throughout the process, as shown in Figure 3d–f.

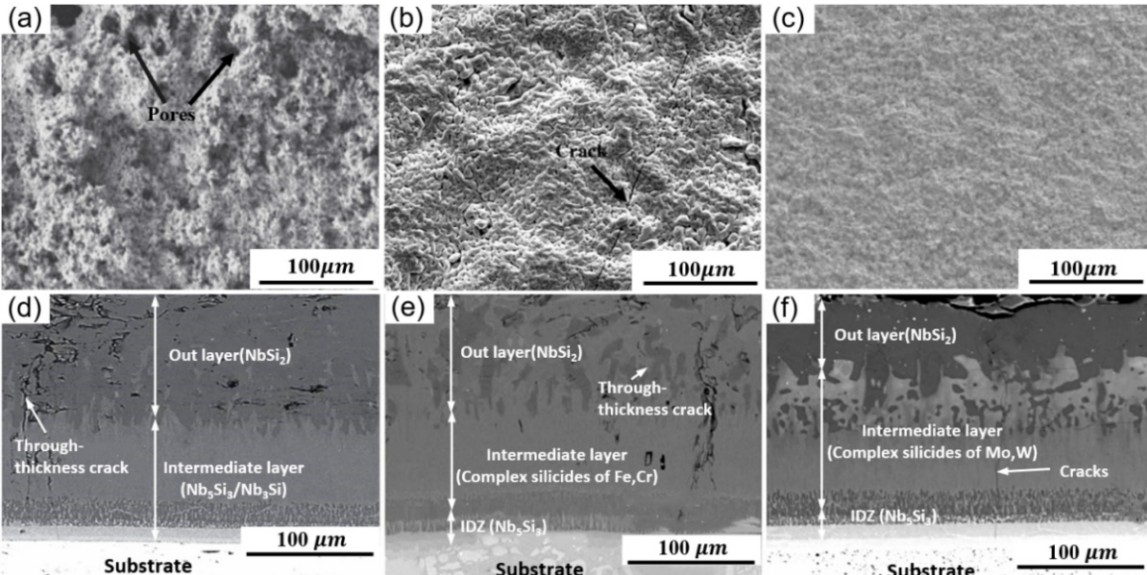

**Figure 2.** Scanning electron microscope (SEM) images of silicide coatings prepared by SS: (**a**,**d**) of pure NbSi$_2$ coating; (**b**,**e**) NbSi$_2$-(Fe, Cr)$_3$Si$_2$ composite coating; (**c**,**f**) NbSi$_2$-(Mo,W)Si$_2$ composite coating.

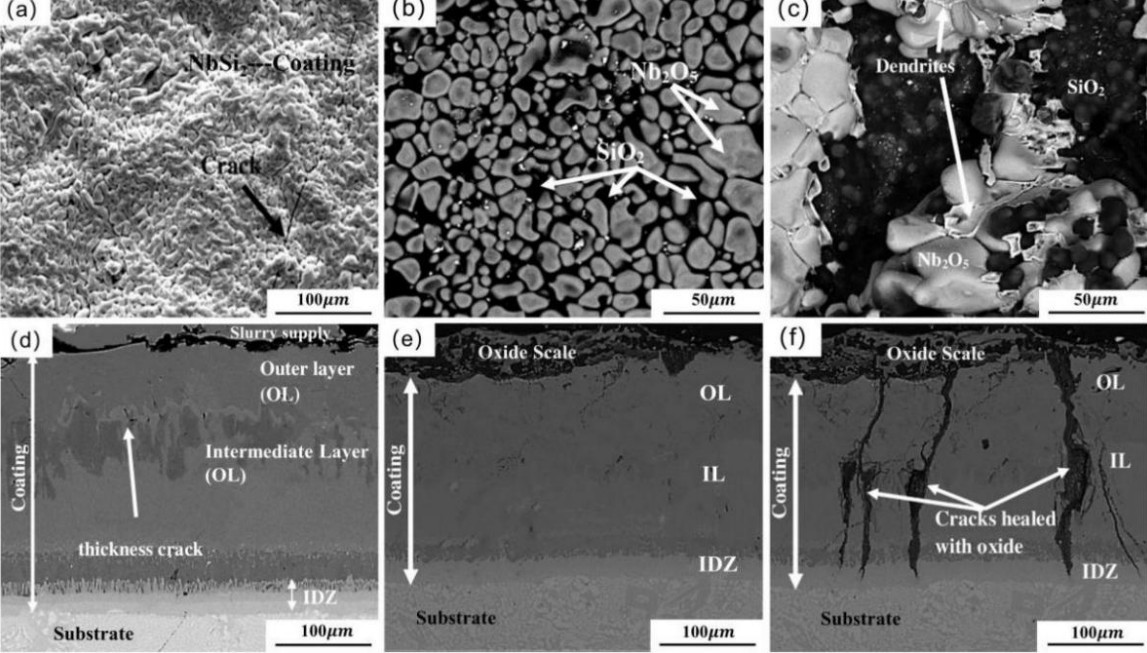

**Figure 3.** SEM images of SS coating before and after oxidation: (**a**,**d**) Before oxidation; (**b**,**e**) After oxidation for 2 h; (**c**,**f**) Coating morphologies after oxidation failure.

## 2.2. Thermal Spraying Method

Thermal spraying is also called sputtering technology. Its principle is to spray molten or semi-molten materials on the surface of the substrate to form a coating through high-speed airflow [28,29]. Due to its high surface coating material, short spraying time, and positive bonding performance between the coating and substrate, it is considered one of the coating preparation technologies with the most development potential [30]. The most typical atmospheric plasma spraying (SPS) coating preparation process is shown in Figure 4. Table 2 summarizes the preparation process, coating structure, and oxidation resistance of SPS coatings on the surface of niobium-based alloys. It can be seen that $MoSi_2$ has been favored by more researchers as the primary spray material [31,32]. This is mainly attributed to its high melting point, good thermal stability, and excellent high temperature creep resistance, which will make the material have a higher melting temperature and longer cooling time when sprayed on the substrate, and significantly improve the bonding strength between the substrate and coating particles. In addition, controlling the material size, carrier gas flow, spray distance, and other parameters will substantially impact the coating quality [33,34].

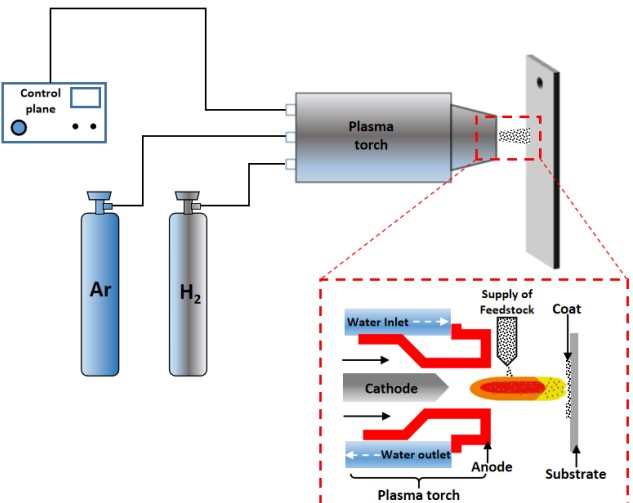

**Figure 4.** Schematic diagram of plasma spraying coating preparation.

The SEM images of typical SPS coatings are listed in Figure 5. The average surface roughness ($R_a$) and porosity of the sprayed Mo coating were the same [35,36]. At the same time, apparent holes and cracks were observed at the cross-sectional coating, as shown in Figure 5a,d. However, with the introduction of other phases such as mullite and $WSi_2$ in the spray material, the surface morphology was significantly improved, and the particle morphology was significantly eased. This was due to the vacancy complementation of the multi-element materials during the spraying process and the self-balance of the thermal conductivity and thermal expansion coefficient during the bonding process, as presented in Figure 5b,c. In addition, a continuous and uniform transition layer was observed between the coating and the substrate, indicating that a positive metallurgical bond was achieved betwixt the substrates and coating, as presented in Figure 5e,f. The author believes that the addition of mullite fills the holes during the spraying process, which can optimize the coating structure, and improves the coating density [37]. The addition of $WSi_2$ can combine with $MoSi_2$ to form $(W, Mo)Si_2$ and $(W, Mo)_5Si_3$, which effectively inhibits the diffusion of Si elements, maintains the coating morphology, and relieves the internal coating caused by the mismatched thermal expansion coefficients defect.

The researchers have studied the oxidation behavior of SPS coatings on Nb-based alloys, as presented in Table 2. The service life of coatings can reach tens to hundreds of hours in an oxidizing environment from 1200 to 1500 °C. The typical SEM of the oxidized coating is shown in Figure 6. It can be perceived that the large oxide particles appeared on

the coating surface after cyclic oxidation at 1500 °C for 43 h. The laser scanning confocal microscopy (LSCM) results show that the $R_a$ of oxidized coating was 23.1 μm, and the main components were $SiO_2$ and $Nb_2O_5$ [38]. Moreover, a large number of holes and cracks were observed at the surface, as shown in Figure 6a–c. This was due to surface defects providing channels for the diffusion of oxygen atoms, resulting in excessive oxidation and expansion inside the coating. Meanwhile, the cyclic thermal shock further aggravated the peeling of the outer oxide protective layer from the inner substrate layer. The cross-sectional image shows that the inside of the oxide layer was very loose with a thickness between 20 to 30 μm, and there was a phenomenon of shedding in its local area, as presented in Figure 6d. However, the mass change of the Mo-MoSi$_2$ coating [39] prepared by Zhang et al. after being oxidized at 1500 °C for 140 h was only $-2.77$ mg·cm$^{-2}$. This is because the addition of mullite inhibits the crystallization of $SiO_2$, improves its fluidity, and promotes the formation of a continuous and dense oxide film on the coating surface [40,41].

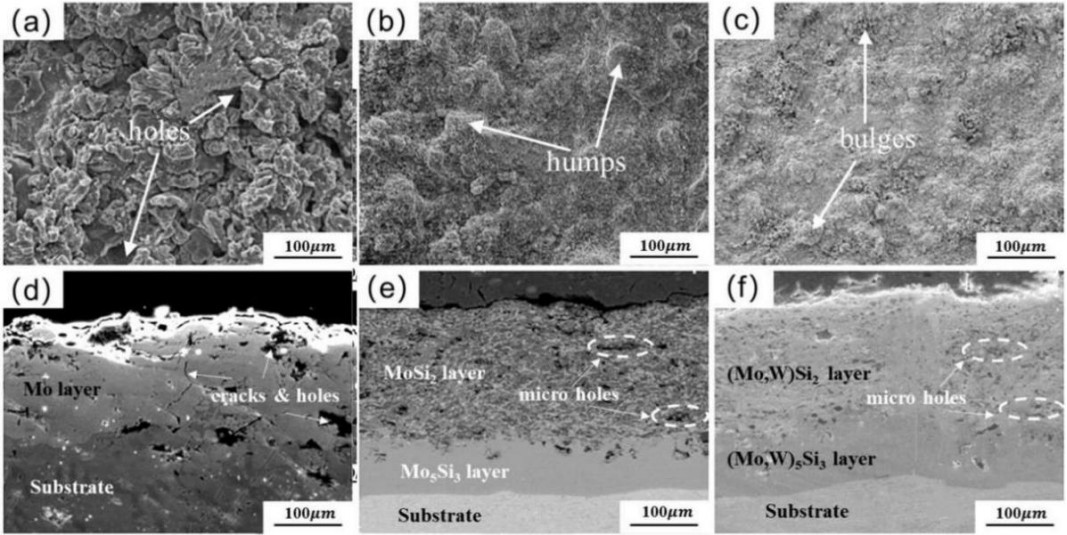

**Figure 5.** SEM images of coatings prepared by SPS under different procedures: (**a**,**d**) Pure Mo coating; (**b**,**e**) MoSi$_2$-mullite coating; (**c**,**f**) WSi$_2$-mullite-MoSi$_2$ coating.

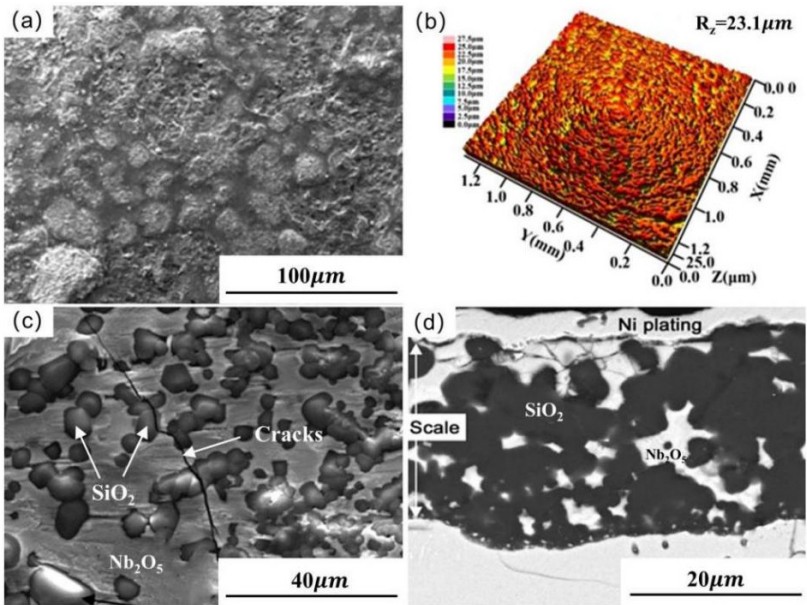

**Figure 6.** SEM and LSCM images of SPS coatings after oxidation at 1500 °C for 43 h. (**a**) Secondary electron image; (**b**) LSCM image; (**c**) Backscattered electron image; (**d**) Cross-sectional images.

**Table 2.** Summary for preparation and oxidation resistance of SPS coatings on Nb-based alloys.

| Substrate | Material Composition (wt.%) | Process Conditions | | Coating Composition and Thickness (μm) | | Oxidation System | Oxidation Products | Quality Change (mg/cm$^2$) | References |
|---|---|---|---|---|---|---|---|---|---|
| | | Spray Distance (mm) | Spray Rate (g/min) | Outer Layer | Inner Layer | | | | |
| Nb521 | 10mullite-90MoSi$_2$ | 100 | 20 | h-MoSi$_2$ t-MoSi$_2$ t-WSi$_2$ (150) | Mo$_5$Si$_3$ W$_5$Si$_3$ NbSi$_2$ (50) | 1500 °C 500 h | SiO$_2$ | 4.41 | [42] |
| Nb521 | MoSi$_2$ 10mullite-90MoSi$_2$ 30mullite-70MoSi$_2$ | 100 | 20 | t-MoSi$_2$ h-MoSi$_2$ (155) | Mo$_5$Si$_3$ Mo (45) | 1500 °C 140 h | SiO$_2$ Mullite SiO$_2$ | 21.38 −2.77 −4.06 | [43] |
| Nb521 | MoSi$_2$ | 100 | 25 | b-MoSi$_2$ (98) | Mo$_5$Si$_3$ (23) | 1200 °C 94 h | MoSi$_2$ | – | [44] |
| Nb-W | MoSi$_2$ | 90 | 20 | h-MoSi$_2$ t-MoSi$_2$ SiO$_2$ (170) | Mo$_5$Si$_3$ (34) | 1500 °C 43 h | SiO$_2$ | 5.31 | [45] |
| Nb-Si-Ti | Mo-45Si-45Al | 90 | 25 | Mo(Si,Al)$_2$ (100) | Mo$_5$(Si,Al)$_3$ (25) | 1250 °C 100 h | SiO$_2$ Al$_2$O$_3$ | 8.24 | [46] |
| Nb-Si-Ti | 2NaF-34Si-B-63Al$_2$O$_3$ | 100 | 25 | MoSi$_2$ (72) | Mo Mo$_5$Si$_3$ (55) | 1250 °C 100 h | SiO$_2$ Borosilicate glass cover | 1.28 | [47] |

*2.3. Embedding Method*

The embedding method is also known as HAPC, and the principle is to place the substrate in a permeation box containing the halide of the coating element and conduct heat treatment in a vacuum or under the condition of continuous inert gas [48,49]. The required coating is formed through vapor migration and reaction–diffusion; the process flowchart is shown in Figure 7. Because this method has the advantages of an uncomplicated process, significant coating efficiency, and freedom from the limitation of the shape of the workpiece, it is widely used in the oxidation protection of refractory metal surface coatings. The researchers conducted a systematic study on the composition and oxidation mechanism of the HAPC coating on the surface of Nb and its alloys, and the results are shown in Table 3. It can be noticed that if the temperature was in the range of 850–1300 °C for 5–25 h, a coating with a thickness of 40–200 μm was obtained. Qiao et al. [50] reported the significance of the growth law of coating crystals on its surface quality. The consequence is that, during the preparation of HAPC coatings, the contact interface between the substrate and the embedding agent undergo chemical combination and diffusion reactions. When the diffusion rate is greater than the reaction rate, many crystal nuclei are formed on the face of the base metal per unit time, which will cause the surface of the base to form a fine and dense coating. Conversely, when the diffusion rate is less than the reaction rate, there are limited crystal nuclei formed on the surface of the base metal per unit time, the formed crystal grains are coarse, the solid phase accumulation is relatively loose, and the coating pore cracks are more severe [51]. The diffusion rate and reaction rate were closely related to the embedding temperature, which theoretically confirms the decisive influence of the embedding temperature. Further research shows that if the thickness of the outer layer of the coating is controlled to 80–100 μm and the inner layer is controlled to 5–20 μm, the resulting coating is denser, with a porosity of about 5–15%.

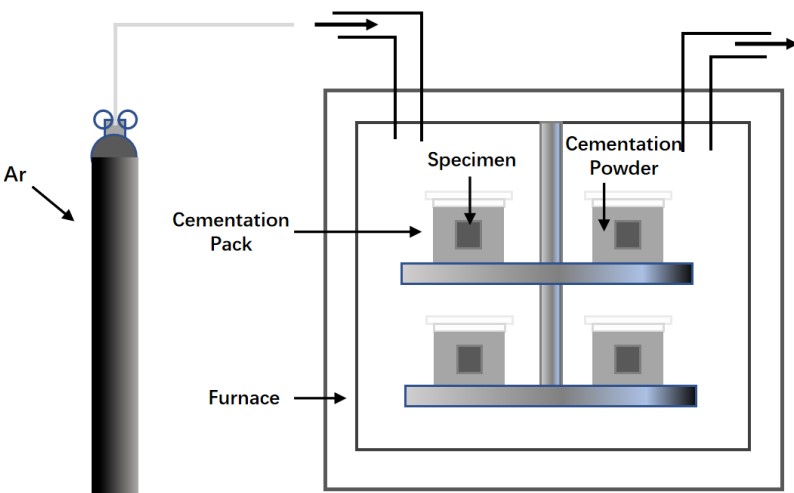

**Figure 7.** Process flowchart of HAPC method.

**Table 3.** Summary for preparation and oxidation resistance of HAPC coatings on Nb-based alloys.

| Substrate | Experimental Conditions | Embedding Components (wt.%) | Coating Composition and Thickness (µm) | | Oxidation System | Oxidation Products | Quality Change (mg/cm$^2$) | References |
|---|---|---|---|---|---|---|---|---|
| | | | Outer Layer | Interface Layer | | | | |
| C-103 | 1100 °C/6 h Ar | 25Si-5NaF-70Al$_2$O$_3$ | NbSi$_2$ Nb$_5$Si$_3$ (60) | Nb$_5$Si$_3$ (5) | 1100 °C/17 h | Nb$_2$O$_5$ SiO$_2$ | 3 | [52] |
| Nb-Cr | 1150 °C/5 h Ar | 10Si-10Al-5NaF-75Al$_2$O$_3$ | Al$_2$O$_3$ Cr$_3$Si Nb$_5$Si$_3$ (35) | Nb$_5$Si$_3$ Cr$_3$Si (5) | 1200 °C/100 h | SiO$_2$ Al$_2$O$_3$ | 3.38 | [53] |
| | 1250 °C/8 h Ar | 10Si-2Y$_2$O$_3$-5NaF-83Al$_2$O$_3$ | (Nb, Cr)Si$_2$ (Cr, Nb)Si$_2$ (185) | (Nb, Cr)$_5$Si$_3$ (15) | 1250 °C/50 h | SiO$_2$ Cr$_2$O$_3$ CrNbO$_4$ | 13.1 | [54] |
| Nb-Si | 1150 °C/10 h Ar | 10Si-10Al-5NaF-2Y$_2$O$_3$-73Al$_2$O$_3$ | (Nb, X)Si$_2$ (Nb, X)$_5$Si$_3$ (40) | (Nb,Ti)$_3$(Al, X) (5) | 1250 °C/50 h | TiO$_2$ Al$_2$O$_3$ SiO$_2$ | 12.5 | [55] |
| | 1300 °C/10 h Ar | 16Si-8Ge-Y$_2$O$_3$-5NaF-70Al$_2$O$_3$ | (Nb, X)(Si, Ge)$_2$ (180) | (Ti, Nb)$_5$(Si, Ge)$_4$ (Nb, X)$_5$(Si, Ge)$_3$ (12) | 1250 °C/100 h | SiO$_2$ GeO$_2$ TiO$_2$ Cr$_2$O$_3$ | 2.78 | [56] |
| Nb-Ti-Al | 850 °C/25 h vacuum | 60Al$_2$O$_3$-40Al | NbAl$_3$ (160) | | 1000 °C/650 h | NbAl$_3$ ($\alpha$-Al$_2$O$_3$) | 1.5 | [57] |
| | 1050 °C/25 h vacuum | 60Al$_2$O$_3$-40Si | NbSi$_2$ (50) | | 1000 °C/650 h | SiO$_2$ TiO$_2$ | 0.4 | |

The typical SEM of HAPC coatings on Nb-based alloys is shown in Figure 8. It can be perceived that the $R_a$ of the Nb-Si-Mo coating [58] was relatively high, and apparent cracks and large granular filler powder were observed at the facial. The internal porosity of the coating was relatively high, consisting of (Nb, X)$_5$Si$_3$ inner layer and (Nb, X)Si$_2$ outer layer, and apparent cracks were observed at the interface layer, owing to the varying grain size of the embedded material [59] and the difference in thermal expansion coefficient between the substrate and the combined coating, as shown in Figure 8a,d. In order to optimize the coating structure, Majumdar et al. prepared a Ge/Ge-Y modified NbSi$_2$ coating on the Nb substrate. It can be perceived that the surface of the Ge modified coating was relatively flat and smooth [60], but a small amount of pore was observed at the grain boundary, as shown in Figure 8b. The interior coating was relatively uniform, but there were still apparent defects. In addition, a large amount of Al and Cr-rich regions were noticed at the junction

of the coating and the substrate, as shown in Figure 8e. Compared with the single-element Ge modified coating, the surface grains of the Ge-Y modified silicide coating after adding Y were smaller, and the average grain size was only 2 to 3 μm, as shown in Figure 8c. It is worth noting that the coating inside was very uniform and dense without any apparent defects [61], and the overall thickness was about 50 μm, as shown in Figure 8f. This is because the addition of the Y element takes advantage of the release of thermal stress and the filling of vacant defects so that the surface grains of the coating are significantly refined, and the gaps between the grains are significantly reduced.

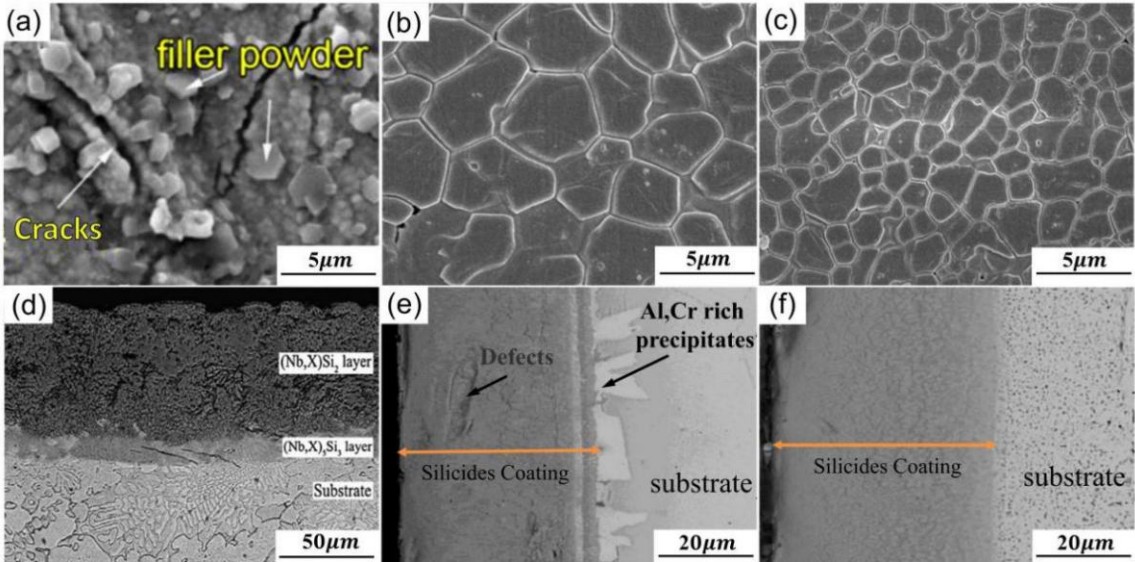

**Figure 8.** SEM of HAPC coatings: (**a**,**d**) Al-NbSi$_2$ coating; (**b**,**c**) Ge/Ge-Y modified coating; (**e**,**f**) Effect of different binders BaF$_2$/CrCl$_3$ on coating quality.

Related researchers have tested the oxidation resistance of HAPC coatings, as shown in Table 3. It can be noticed that the mass gain of the coating was only 0.4–13.1 mg·cm$^{-2}$ after being oxidized at 1000–1250 °C for 17–650 h. Typical SEM of oxidized coating is shown in Figure 9. Overall, the oxidized Nb-Si-Ti coating surface was smoother, covering a layer of molten oxide film, but a few holes were still observed [62]. The author believes that this is related to the higher surface roughness of the coating before oxidation, resulting in uneven oxidation and the volatilization of Nb$_2$O$_5$ during the oxidation process, as shown in Figure 9a. The oxidation results show that the main component of the oxide film was SiO$_2$-TiO$_2$, and the thickness was about 20 μm. In addition, a small amount of Cr$_2$O$_3$ was also observed at the bottom of the oxide layer [63], as presented in Figure 9c. However, the oxidized (Nb, X)Si$_2$ coating surface was very rough, and a large number of oxide particles mainly composed of SiO$_2$-Al$_2$O$_3$ were observed, as shown in Figure 9b. A large number of holes and longitudinal cracks across the entire cross-section appeared in the coating. At the same time, partial areas of the oxide layer appeared to fall off [64]. It is worth emphasizing that an extended layer of Cr$_2$O$_3$ was observed betwixt the oxide layer and the internal coating, which relieved the further oxidation of the coating to a certain extent, as shown in Figure 9d.

## 2.4. Other Methods

By analyzing the advantages and disadvantages of the coating preparation process, related scholars appropriately combined different methods to make up for each other's advantages and disadvantages, thereby obtaining a composite coating with superior oxidation resistance [65,66]. Common combination categories, process conditions, coating composition, thickness, and oxidation characteristics are shown in Table 4. It can be seen that the coating system was dominated by Mo-Si-X (B, Ce, etc.), which is related to the high

thermal conductivity and good thermal stability of $MoSi_2$. The addition of elements such as B and Ce can further optimize the coating structure and improve its oxidation resistance at high temperatures [67]. The thickness of the coating prepared by the two-step process was 55 to 160 μm. The outer layer mainly consisted of high-priced metal and high silicide, such as $MoSi_2$. Due to the difference in the concentration of internal and external elements, the inner layer was mostly $(Nb, X)_5Si_3$ and $(Nb, X)_3Si$. The typical coating surface and the corresponding cross-sectional morphology are shown in Figure 10. Since most of the later steps of this kind of method are HAPC, the coating surface exhibited a relatively high toughness, and a large amount of granular embedded powder was observed to adhere, as shown in Figure 10a,c. The main component of the coating layer was a mixture of $MoSi_2$ and the second phase was loosely bonded and had many holes and cracks [68,69]. The inner layer mainly consisted of Nb-based metal silicide; its cross-sectional morphology was good, and the structure was dense and uniform. This shows that the coating and the substrate achieved good metallurgical bonding, as shown in Figure 10b,d. This type of coating preparation method realizes the diffusion and filling of the outer layer components to the inner layer components through the latter process so that the structure of the inner coating is further strengthened, and the use performance of the coating is significantly improved. In addition, technologies such as laser cladding technology (LCT), physical vapor deposition (PVD), and hot-dipped silicon (HDS) [70] have also been favored by related scholars in the preparation of Nb surface anti-oxidation coatings. Zhang et al. used HDS [71,72] technology to prepare a $WSi_2$ coating [73,74] with nano-level roughness on the surface of W. The resulting coating facial was absolutely dense and uniform without defects such as holes, gaps [75], etc. This will help the technology apply the coating to the surface of Nb and its alloys, and provides a helpful reference for the preparation of anti-oxidation coating.

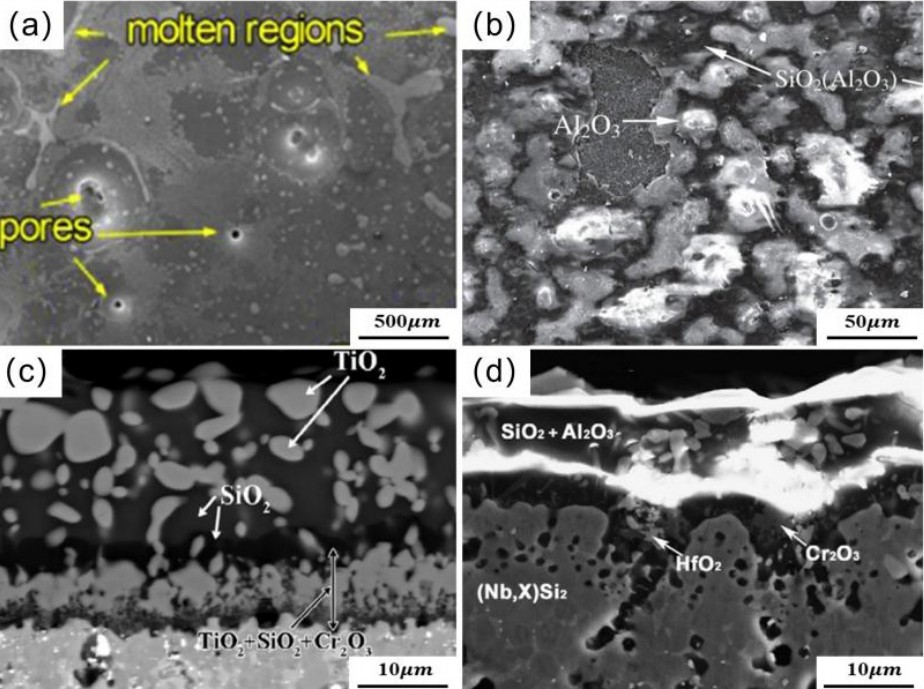

**Figure 9.** SEM of HAPC coatings after oxidation. (**a**,**c**) Surface and cross-sectional morphology of Nb-Si-Ti coating after oxidation at 1050 °C for 5 h; (**b**,**d**) (Nb, X)$Si_2$ coating surface and cross-section morphology after oxidation at 1250 °C for 50 h.

**Table 4.** Two-step coating preparation procedure and oxidation resistance details on Nb alloy surface.

| Substrate | Process Conditions | | Coating System | Coating Composition and Thickness (μm) | | Oxidation System | | Oxidation Products | Quality Change (mg/cm²) | References |
|---|---|---|---|---|---|---|---|---|---|---|
| | | | | Outer Layer | Interface Layer | Oxidation Temperature (°C) | Oxidation Time (h) | | | |
| Nb | SPS SD: 100 mm SR: 20 g/min | HAPC Ar 1000 °C/50 h | Mo-Si–B | $MoSi_2$ $NbB_2$ $NbSi_2$ (70) | $Nb_5Si_3$ $NbB_2$ (10) | 1300 | 24 | $Nb_2O_5$ $SiO_2$ | 0.44 | [76] |
| C-103 | HAPC 1100 °C/6 h | HAPC 1050 °C/4 h | Si-B | $NbSi_2$ $NbB_2$ (125) | $NbB_2$ (13) | 1300 | 100 | $Nb_2O_5$ $NbO_2$ $B_2O_3$ | 1.44 | [77] |
| Nb-Si | SS 1550 °C/2 h | HAPC Ar 1200 °C/5 h | Mo-Si-Ce | $NbSi_2$ $MoSi_2$ (80) | $Nb_5Si_3$ (4) | 1600 | 24.7 | $SiO_2$ | 3.57 | [78] |
| Nb-Si-Ti | PVD 300 °C/2 h | HAPC Ar 1450 °C/12 h | Mo-Si–B | $MoSi_2$ $(Nb,Ti)_5SiB_2$ (50) | $(Nb, X)_5Si_3$ (5) | 1300 | 24 | $MoO3$ $SiO_2$ | −0.55 | [79] |
| Nb-Si-Ti | SPS SD: 60 mm SR: 90 g/min | HAPC Ar 1250 °C/4 h | Si-Y-Zr | $(Nb,Ti)_5Si_4$ (110) | $(Nb, X)_5Si_3$ (5) | 1250 | 100 | $Nb_2O_5$ $TiO2$ $SiO_2$ | 1.6 | [80] |
| Nb-Si-Ti | SPS SD: 90 mm SR: 20 g/min | HAPC 1000 °C/40 h | Mo-Si-B | $MoSi_2$ $MoB$ (115) | $Mo$ (45) | 1250 | 100 | $B_2O_3$ $SiO_2$ | 0.92 | [81] |

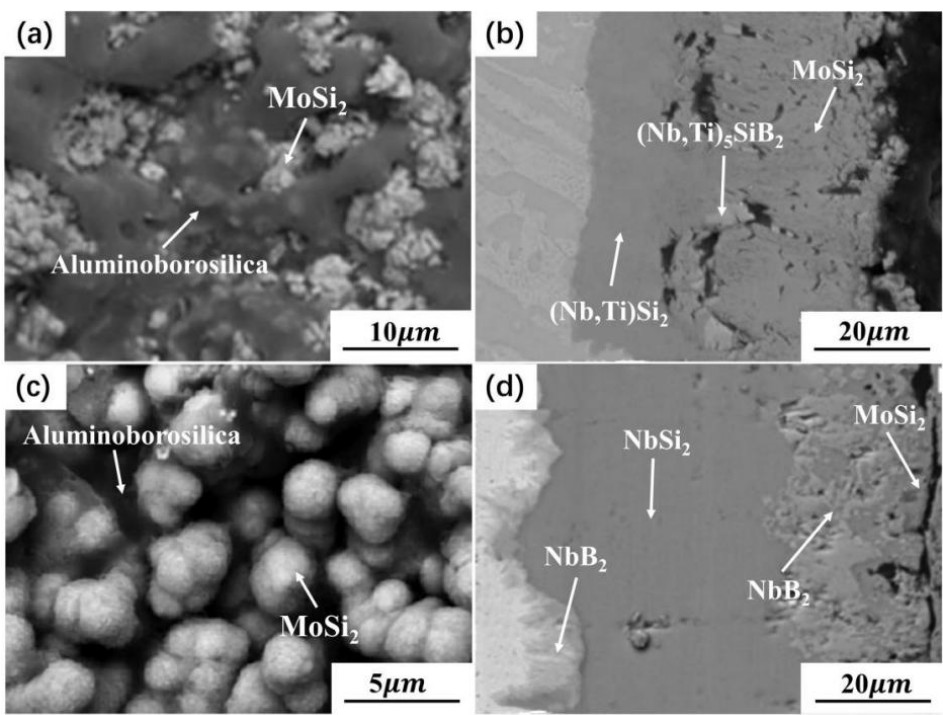

**Figure 10.** Typical SEM of two-step method coatings. (**a,b**) Mo-Si-B coating prepared by SPS (spray distance:100 mm, spray rate: 20 g/min) + HAPC (1000 °C/50 h) process; (**c,d**) Mo-Si-B coating prepared by PVD (300 °C/2 h) + HAPC (1450 °C/12 h) process.

Figure 11 present the typical SEM of oxidized coatings prepared by the two-step approach. The mass loss per unit area of the coating was within 4 mg·cm$^{-2}$ after being oxidized in the range of 1300–1600 °C for 24–100 h. The outward layer of the oxidized coating was approximately flat and smooth without apparent cracks and holes. This was due to the formation of aluminoborosilica with a certain fluidity during the oxidation process, filling in the surface defects [82], as shown in Figure 11a,c. $MoSi_2$-$SiO_2$ dominated the outer phase of the oxidized coating. Due to original defects and continuous consumption during the oxidation process, the partial area of the oxide layer fell off, showing discontinuity

and inhomogeneity [83]. However, the inner layer organization of the coating was still dense and compact without apparent flaw, and the transition layer of $(Nb, X)_5Si_3$ and $NbB_2$ was observed to grow inward at the interface between it and the substrate, as shown in Figure 11b,d.

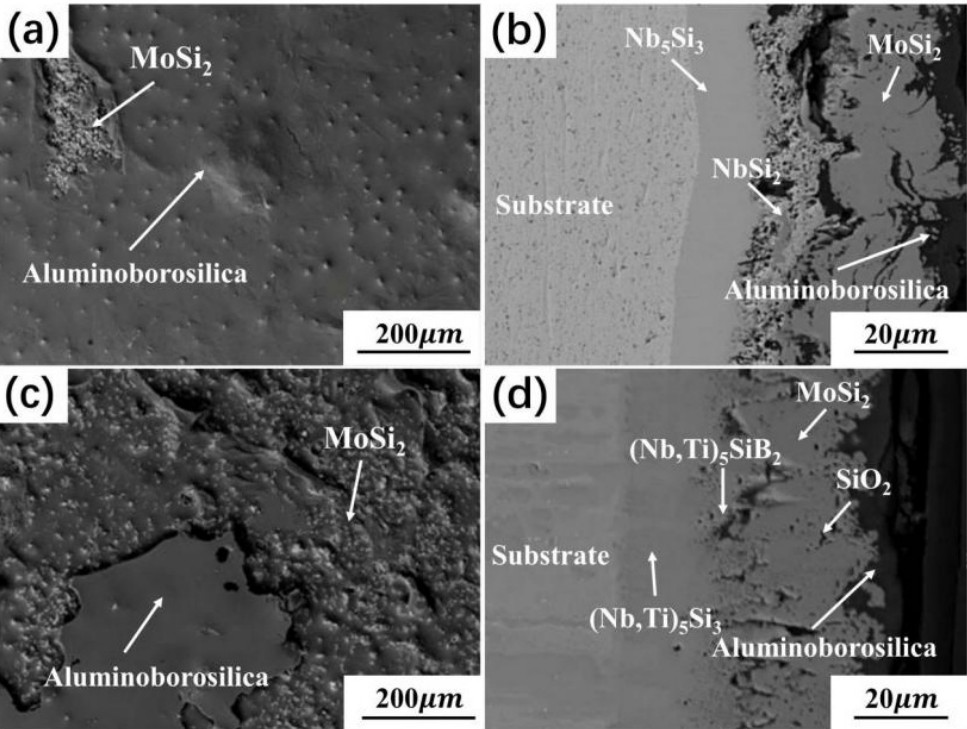

**Figure 11.** SEM of oxidized coatings prepared by the two-step method, (**a**,**b**) Mo-Si-B coating after oxidation at 1250 °C for 100 h; (**c**,**d**) Mo-Si-B coating after oxidation at 1300 °C for 24 h.

### 3. Oxidation Mechanism and Failure Behavior of Coating

According to the summary of the oxidation characteristics of Nb alloys surface coatings, the oxidation behavior and failure mechanism are summarized, as shown in Figure 12. It can be noticed that oxidation can be divided into two stages. The inner and outer layers of the coating are composed of the $Nb_5Si_3$ layer and $NbSi_2$ layer, respectively, as shown in Figure 12a. In the initial stage of oxidation, the oxygen-philic compounds on the coating surface are rapidly oxidized. The oxidation reaction is more severe at weak areas such as cracks, gaps, etc., and the generated oxides such as $Nb_2O_5$ and $SiO_2$ are transformed into defects, start to grow, and gradually spread to the entire surface. As the reaction progresses, $Nb_2O_5$, $NbO_2$, etc., gradually volatilize, leaving many holes on the surface. In addition, due to the release of thermal stress and the mismatch of thermal expansion coefficients between systems, many cracks sprout on the surface of the coating. At the same time, the addition of some modifying elements (X) improves the fluidity of $SiO_2$, fills up these defects to a certain extent, and forms an $Nb_2O_5$-$SiO_2$-$X_2O_3$ protective film system on its surface, as shown in Figure 12b. At last, the oxide layer gradually thickens, the $NbSi_2$ layer as the central part of the coating is gradually consumed, and the self-healing ability of the coating gradually deteriorates. However, the $Nb_5Si_3$ layer with poor oxidation resistance gradually becomes thicker. With the oxidation process, the low oxidation resistance $Nb_5Si_3$ layer is gradually destroyed, resulting in the oxidation failure of the coating, and a large number of holes and cracks are observed, as shown in Figure 12c.

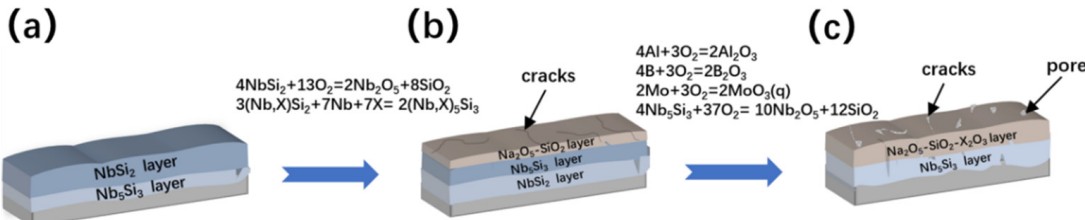

**Figure 12.** Oxidation and failure mechanism of silicide coatings on Nb-based alloys, (**a**) Before coating oxidation; (**b**)Primary oxidation of coating; (**c**) Deep oxidation of coating.

## 4. Conclusions and Prospects

In this work, the preparation methods of anti-oxidation coatings on Nb-based alloys are reviewed, and the structure and anti-oxidation performance of coatings obtained by different methods are summarized, as shown in Figure 13. Overall, the high-temperature oxidation resistance of Nb-based alloys has been significantly enhanced by surface coating technology. Through in-depth comparison and analysis of various methods, it can be known that the volatilization of solvents and cement, and the uneven particle size of the mixture during the sintering process, result in poor surface quality and high porosity of the coating prepared by SS. Although the two processes of HAPC and CVD have no volatilization phenomenon and are not limited by the shape of the substrate, their lower deposition temperature makes the growth of the coating slower, and the preparation cycle is longer. In contrast, due to its high diffusion temperature, SPS can deposit coatings of tens to hundreds of microns in a short time. However, due to the uneven melting of the spray paint and a small amount of gas during the spraying process, the porosity of the coating is higher, and the bond with the substrate is poor. In addition, although the two-step coating has a relatively excellent structure, its process is complicated, and the coating preparation efficiency is low. As a new coating preparation process, HDS technology dominates due to short deposition time, high coating preparation productivity, smooth and dense coating surface, etc., and it is expected to protect Nb-based alloys at high temperatures.

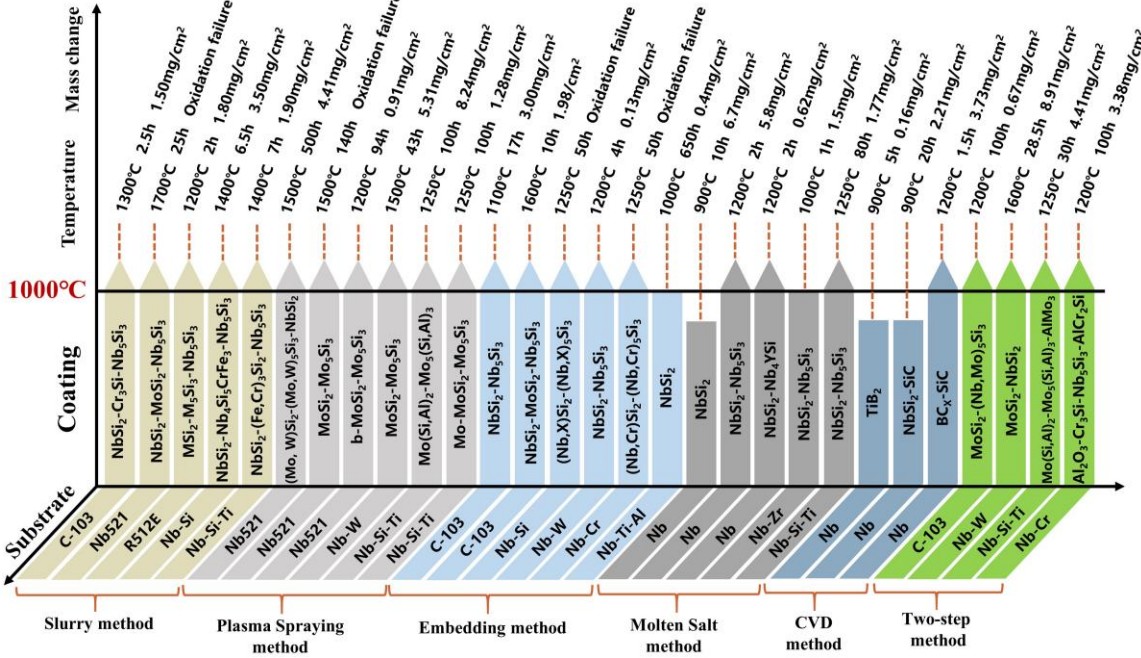

**Figure 13.** Overview of preparation parameters and oxidation resistance of Nb-based alloys surface coatings.

Summarizing the anti-oxidation mechanism of the coating prepared by the above method, it can be found that the outer protective scale of the coating with better anti-oxidation performance is generally composed of $SiO_2$ and other inert melt film shielding substances. However, inert molten films such as $SiO_2$ are formed after oxidation tests, and the oxidation process is challenging to control. Therefore, in order to further improve the high-temperature oxidation resistance of the coating, some beneficial elements are usually added in an appropriate amount during the coating preparation process. Among them, the "selective oxidation type" alloy element X (X = A1, Cr, Mo, Ti, etc.) is added to make it preferentially combine with the O element to form an oxide during the high-temperature oxidation process, setting a dense isolated layer on the surface of the substrate. This blocks the inward diffusion of oxygen atoms, inhibits the formation of $Nb_2O_5$, and reduces the oxidation rate. The addition of element B and mullite can improve the fluidity of $SiO_2$ and promote the formation of a uniform and thick oxide film on the coating surface. The addition of Y and Ce elements can refine the coating grains, optimize the coating structure, and significantly improve the strength of the coating at high temperatures so that it can maintain a good shape during the oxidation process. The addition of W and Ge can constrain the diffusion of Si elements into the substrate, slow down the generation of $Nb_5Si_3$ with poor oxidation resistance, and lengthen the oxidation service life of the coating. The introduction of a proper amount of mullite can fill the pores inside and on the coating surface, optimize the coating structure, increase the density of the coating, inhibit the recrystallization of $SiO_2$, and promote the thick oxide film on the surface of the coating. In addition, optimizing the coating preparation process and structure can significantly reduce defects produced by thermal expansion coefficient mismatch between coating and substrate, which also plays a crucial role in improving the high-temperature oxidation resistance of the coating.

**Author Contributions:** Conceptualization, Y.Z.; methodology, Y.Z., X.Z. and T.F.; validation, Y.Z. and K.C.; formal analysis, J.W. and F.S.; investigation, L.Y., H.M. and T.F.; resources, Y.Z.; data curation, Y.Z., X.Z. and T.F.; writing—original draft preparation, Y.Z., X.Z. and T.F.; writing—review and editing, Y.Z., X.Z. and T.F.; visualization, X.Z. and T.F.; supervision, Y.Z.; project administration, Y.Z.; funding acquisition, Y.Z. All authors have read and agreed to the published version of the manuscript.

**Funding:** This research was funded by the National Natural Science Foundation (51604049).

**Institutional Review Board Statement:** Not applicable.

**Informed Consent Statement:** Not applicable.

**Data Availability Statement:** Not applicable.

**Acknowledgments:** The authors wish to acknowledge the contributions of associates and colleagues at Anhui University of Technology. The financial support of the National Natural Science Foundation.

**Conflicts of Interest:** The authors declare no conflict of interest.

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
