# Peer review of "The Protection, Challenge, and Prospect of Anti-Oxidation Coating on the Surface of Niobium Alloy"

_coatings, doi:10.3390/coatings11070742_

Round 1
Reviewer 1 Report
overall quality of the work is good only thing kindly concentrate on language editing
Author Response
Response to Reviewer 1 Comments
Point 1: Overall quality of the work is good only thing kindly concentrate on language editing.
Response 1: Thank you very much for taking the time to review this manuscript and put forward valuable opinions on the problems in it. According to your suggestion, the accuracy of language narration and text format will be improved again by improving the language narration of manuscripts; For example, in line 236, "mass change" was replaced by "mass gain"; Re-labeling of cathode and anode spelling in Fig 4, etc.
Reviewer 2 Report
Dear Authors,
Thank you for making the effort at putting together so much scattered information on Nb anti-oxidation coatings into a synthetic summary. The detailed comments, not too many, are given in the file attached.
One general comment has to do with the possible difference between the oxidation testing conditions (a typical laboratory practice) and the intended conditions at the target applications. Oxidation testing is performed in still atmosphere and at no externally induced mechanical stress. If the coating is to be used on parts subjected to friction (even against air) or pressure the coatings' performance may be compromised due to mechanical damage of the coating and/or its bonding to the substrate. The problem is worth a mention.
Regards
Your reviewer

Author Response
Response to Reviewer 2 Comments
Point 1: It might be interesting, provided such data gets published, to attempt comparing different coatings with regard to their strength of bonding to the substrate. It is obvious that a typical oxidation test is carried out on specimens surrounded by atmosphere only with only rudimentary contact with solids (just enough to provide support), but several methods are commonly used to check the bond between the coating and the substrate (e.g. scratch testing). If currently such data is unavailable, perhaps it can be an interesting thread in the research on anti - oxidation coatings?After all, there is some mention in the initial part of this paper of high mechanical loads applied to the external surfaces of space shuttle vehicles upon atmospheric reentry [6]. A weakly bonded coating is likely to be removed under such conditions thus exposing the substrate to a high temperature oxidizing environment.
Response 1: Your suggestion is essential to me, and the manuscript can be further improved by comparing the bonding strength between the coating and the substrate. As you said, for protective coatings, thermal shock performance and oxidation resistance are essential indicators to judge the versatility of protective coatings. At present, the references tend to focus more on the oxidation resistance of coatings, and the description of bonding properties is relatively few, which makes it challenging to summarize the high-temperature thermal shock properties of different coatings through this manuscript. We will pay attention to your proposal and will focus on the test of adhesion strength and thermal shock performance between coating and substrate in the future.
Point 2: Please consider using the notion of 'mass gain' or 'mass loss' instead. Of course, a positive change in mass is a mass gain and the negative change in mass is mass loss, but for the sake of ease of reading the proposed change seems appropriate.
Response 2: Thanks a lot for pointing out that the accuracy of the language needs to be standardized. As you said, we need to change the 'mass change' here to the 'mass gain'.
Point 3: This infographic figure is an interesting concept of presenting the various results jointly. Apart from numerical values indicating the kinetics of oxidation, oxidation temperature and time figures are given. However difficult it might be to do, an attempt should have been made at bringing the results to a common denominator, at least in terms of the duration of the test. The spread of test times ranging from 1 h to 650 h makes it difficult to comprehend the significance of the message being presented. Oxidation curves are usually steep in the initial part of the oxidation test and have a tendency to level out further into the test. Would that be possible to extract such data from the reviewed batch of research papers?
The comparison presented would be much clearer if the performance of each coating was somehow emphasized e.g. by colour code. The simplest proposition being: green for good, yellow for mediocre, red for unacceptable or failed.
Response 3: Thank you very much for asking whether the oxidation resistance of the coating can be summarized by the length of oxidation resistance time in Figure 13. We have previously tried to evaluate its antioxidant activity by comparing its antioxidant duration; However, through reading many kinds of literature, we find that the effect of oxidation temperature on the oxidation resistance of the coating is far more significant than that of oxidation time. In addition, the whole oxidation process is rarely recorded in detail in the relevant literature, which makes it difficult to accurately define its oxidation resistance at different oxidation temperatures and different oxidation times. For your proposal that its antioxidant performance can be distinguished according to color, it is of great guiding significance to us; We initially divided the base metal by its complexity and the preparation method of the coating; If you adopt your proposal, it will break the original division rules.
Reviewer 3 Report
- I recommend inserting a few introductory sentences between lines 66 and 67.
- Table 1 - Think about the adjustment, the table looks a bit confusing.
- All pictures - are they created by the author or taken from literature? This fact should be obvious.
- In the text, resources are sometimes placed just after the word without a space - for example: line 51 or 74, go through the text carefully and correct it.
- On the line 85 is "(a)and(b) - insert spaces.
- See line 115 - a similar problem, see above.
- See line 119 - a similar problem, see above, go through the text carefully and correct it.
- It would be good to think about the distribution of text and figures, because there are now big blanks in the document, which is not good for the overall formal tone - pp. 3, 4, 5, 7, 9, 10, 15
- It's not good for the figure caption to be on the other side - fig. 5, 10
- The text format on lines 294-303 is weird - need to change the format.
- Some acronyms in the text are not explained, it is necessary to add this (eg HDS).
The article provides a relatively detailed and interesting summary of the topic. But it is only a summary of the current state and there is some outline of future possibilities (prospects). It does not deal with any specific experiment (research) and its evaluation. However, if this fact does not conflict with the principles of publication in the journal, I have no objections.
Author Response
Response to Reviewer 3 Comments
Point 1: I recommend inserting a few introductory sentences between lines 66 and 67.
Response 1: Your suggestion is of great guiding significance to us. We insert a narrative introduction between lines 66-67 to transition before and after the article. The revised version will be given in the attachment.
Point 2: Table 1 - Think about the adjustment, the table looks a bit confusing.
Response 2: Thank you very much for pointing out whether Table 1 can be partially deleted and adjusted. We have tried to omit the line "slurry composition" before but considering that omitting this part will make readers unable to know the slurry composition and the proportion of each component. In addition, other information shown in Table 1 will be necessary to readers, so it is not easy to modify and delete the table.
Point 3: All pictures - are they created by the author or taken from literature? This fact should be obvious.
Response 3: All pictures are taken from references and marked in the reference column in the table above the pictures. In order to avoid repeated citation of the same references, the annotation of picture information is omitted.
Point 4~11: In the text, resources are sometimes placed just after the word without space - for example, line 51 or 74, go through the text carefully and correct it. On the line 85 is "(a)and(b) - insert spaces. See line 115 - a similar problem, see above. See line 119 - a similar problem, see above, go through the text carefully and correct it. It would be good to think about the distribution of text and figures because there are now big blanks in the document, which is not good for the overall formal tone - pp. 3, 4, 5, 7, 9, 10, 15. It's not good for the figure caption to be on the other side - fig. 5, 10. The text format on lines 294-303 is weird - need to change the format. Some acronyms in the text are not explained, it is necessary to add this (eg HDS).
Response 4~11: Thank you very much for your above modification to the text format of this manuscript. We have carefully read your modification suggestions and have revised the text format you pointed out. The revised manuscript can be found in the annexe.
Reviewer 4 Report
good paper on the protection, challenge, and antioxidation coating of niobium alloy.
Introduction and anti-oxidation coating on Nb alloys covers sufficient information about the research area:
in line 124: can you comment more on the principle of thermal diffusion.
section 3 oxidation mechanism and failure behavior of coating: very well written and explained.
Author Response
Point 1: Please comment on "thermal diffusion" in line 124.
Response 1: Thank you very much for taking the time to review this manuscript and put forward valuable opinions on the problems in it. In line 124, you proposed, we explained "thermal diffusion" as follows: Thermal difference referred to the spontaneous transition of matter to an equal state in a high temperature and closed environment, significantly affected by temperature. Generally speaking, when the temperature rates, the average kinematic energy of molecules will increase, and more molecules will achieve "activation," which will improve the combination efficiency and promote the transition process to an equal state. In this paper, after inquiring about the material balance trend through the "Material Project Reaction Calculator," we know that the binary combination of Nb, W, and other metals with Si has a more significant trend to produce (Nb,X)5Si3, The intermediate products such as (Nb,X)Si2 will spontaneously transform into (Nb,X)5Si3 In a high temperature and closed environment. In a short time, the transition layer of (Nb,X)5Si3 growth thicker. With the passage of bonding time, the whole coating will be transformed into (Nb,X)5Si3 final state. The brief version of this note has been modified in the original manuscript according to your suggestion